# The Theoretical Study of Kink Deformation in Graphite Based on Differential Geometric Method

**DOI:** 10.3390/nano12060903

**Published:** 2022-03-09

**Authors:** Xiao-Wen Lei, Shungo Shimizu, Jin-Xing Shi

**Affiliations:** 1Department of Mechanical Engineering, University of Fukui, 3-9-1 Bunkyo, Fukui 910-8507, Japan; shimishun41@gmail.com; 2Precursory Research for Embryonic Science and Technology (PRESTO), Japan Science and Technology Agency (JST), Saitama 332-0012, Japan; 3Department of Production Systems Engineering and Sciences, Komatsu University, Nu 1-3 Shicyomachi, Komatsu 923-8511, Japan; jinxing.shi@komatsu-u.ac.jp

**Keywords:** graphite, kink deformation, mean curvature, laminated structure, differential geometry

## Abstract

Kink deformation is often observed in materials with laminated layers. Graphite composed of stacked graphene layers has the unique laminated structure of carbon nanomaterials. In this study, we performed the interlayer deformation of graphite under compression using a simulation of molecular dynamics and proposed a differential geometrical method to evaluate the kink deformation. We employed “mean curvature” for the representativeness of the geometrical properties to explore the mechanism of kink deformation and the mechanical behaviors of graphite in nanoscale. The effect of the number of graphene layers and the lattice chirality of each graphene layer on kink deformation and stress–strain diagrams of compressed graphite are discussed in detail. The results showed that kink deformation occurred in compressed graphite when the strain was approximately equal to 0.02, and the potential energy of the compressed graphite proportionately increased with the increasing compressive strain. The proposed differential geometric method can not only be applied to kink deformation in nanoscale graphite, but could also be extended to solving and predicting interlayer deformation that occurs in micro- and macro-scale material structures with laminated layers.

## 1. Introduction

Graphite that is composed of stacked graphene layers has the unique laminated structure of carbon nanomaterials. Although graphene that consists of a hexagonal pattern of carbon atoms has extremely excellent mechanical behavior, natural graphite with laminated graphene is soft and easy to deform under high pressure due to the weak van der Waals (vdW) interaction force between the adjacent graphene layers [1]. When deformation occurs in graphite, its mechanical properties may be changed due to its laminated structure. Moreover, exploring the deformation behavior of graphite not only investigates the mechanical behavior of graphite itself, but it also establishes a general theory for studying other materials with similar laminated structures. Ref. [2] emphasized that “The fact that one can learn something about an earthquake from studying the deformation of graphite, is quite astonishing and remarkable indeed”. Kink deformation in bending graphite was observed from experimentation, and it was studied by theoretical analysis and molecular dynamic (MD) simulation [3]. Band theory [4] and discrete-plate theory [5] were employed to investigate the kinking phenomenon in graphite. The stability, elastic properties, and deformation behavior of graphene-based, diamond-like phases were analyzed by MD simulation [6], where Poisson’s ratio of graphene with the lattice defect was identified [7].

According to the unique deformation phenomena in laminated structures, the concept of the kink band was proposed by [8], where the generation of a kink band that would produce kink deformation was studied theoretically. Ref. [9] predicted the kink band angle and compressive strength of materials with laminated structures. Ref. [10] performed continuum mechanical modeling of the formation of kink bands in fiber-reinforced composites, from where the kink propagation stress, the kink orientation angle, and the fiber direction within the kink band could be determined completely. Compressive failure due to steady state kink-band broadening in materials with laminated structures was analyzed theoretically by [11], where an incremental scheme was proposed for calculating kink-band broadening stress and the lock-up conditions of the kink band. Ref. [12] proposed a new micromechanical model of the broadening of kink bands in fiber-reinforced composites, which was employed to simulate the steady-state axial propagation of kink bands. Ref. [13] described the mechanisms of kink-band broadening in fiber-reinforced composites using the softest deformation modes and measured kink-band broadening stress using experiments. Refs. [14,15] calculated the critical strain of fiber-reinforced composites; he found that fibers could break in the first step and a fully developed kink band could exist in the second step. Using micromechanical models, Vogler et al. [16] performed theoretical analysis on the initiation and the growth of kink bands in fiber-reinforced composites, which was motived by experimental findings. The Maxwell stability criterion was also proposed to investigate the properties of kink bands [17,18]. Microscopic deformation in structures caused by crystal plasticity were studied using theoretical analysis [19,20] or computational approaches [21,22]; the simplified explanation of the mechanism of a kink band would be also useful for the coarse-grained model. Kink deformation played an important role in achieving a reversible hysteresis in a pseudo elastic deformation [23,24]. The compression behavior of natural graphite sheet [25] has been reported, and it points out that deformation is significant during the forming process. The determination of graphite deformation behaviour using microtribological pressure tests [26] has been investigated using a numerical approach. Moreover, ref. [27] have discussed the characterization of ripplocation mobility in compressive graphite. In addition, Ref. [28] proposed the nonlinear continuum theory to describe the buckling behavior of graphene and Ref. [29] discussed the phonon frequency of graphite under uniaxial compression along the *c* axis. However, the deformation of graphite has not been described by exact quantitative figures, and there is no explanation of the relationship between the shape of it and force. Especially, the deformation mechanism of laminated structures was still not clarified clearly enough up until now.

The establishment of a general deformation theory based on a differential geometric method is expected to elucidate the deformation mechanism in laminated structures, e.g., graphite in the present work. To investigate kink deformation in micro-buckled graphite with laminated graphene layers, in the following parts, we build two kinds of atomic simulation models: armchair graphite and zigzag graphite. Both kinds of graphite incorporate 20, 30, 40, 50, or 60 atomic layers, and we performed the compression test on each type of graphite using MD simulation at first. Then, we proposed a differential geometric method to evaluate the mean curvature of each graphene layer in each type of graphite. Next, we discussed the geometrical properties of unique deformation and studied the relationship between the distributions of mean curvature and potential energy. Lastly, we summarized the remarkable conclusions.

## 2. Atomic Model of Graphite with Layered Structure

In this study, we built an atomic model of graphite composed of laminated graphene layers where each layer of graphene consisted of perfect hexagonal lattice of carbon atoms, which is shown in Figure 1. The atomic model of graphene was located on the x−y plane, whose dimensions in *x* direction ($l_{x}$) and *y* direction ($l_{y}$) were 102 Å and 49 Å, respectively. Moreover, the heights of models were dependent on the number of graphene sheet (GS) layers. We noted here that the dimension of a simulation model was representative for clearly illustrating the mechanism of kink deformation in graphite.

We set the lamination direction along the *z* direction with the well-known ABAB stacking planar arrangement. The number of layers *n* in each simulation were set as 20, 30, 40, 50, and 60, respectively. Moreover, the layers were numbered from bottom to top with the initial interlayer distance of 3.35 Å [30]. According to the chirality of carbon nanomaterials, we modeled two kinds of graphite, i.e., armchair and zigzag, and applied the compression load along both the armchair direction and the zigzag direction.

## 3. Analysis Method

### 3.1. Molecular Dynamics Simulation

According to the atomic model, we performed the compression test to elucidate the mechanism of kink deformation in graphite by MD simulation using the large-scale atomic/molecular massively parallel simulator (LAMMPS). The interaction was evaluated using the adaptive intermolecular reactive empirical bond-order (AIREBO) potential [31], where the cut-off distance between carbon atoms $r_{\mathrm{cut}}$ was set to 1.95 Å. The AIREBO potential includes three potential terms: the first term is the reactive empirical bond-order (REBO) potential of Brenner, which describes the interactions between carbon atoms in each graphene layer; the second term is the Lennard Jones (LJ) potential, which describes the interaction between the neighboring layers and is evaluated by the vdW interaction force [32]; the third term is the explicit four-body potential, which describes various dihedral angles caused by obvious out-of-plane deformation. To make an energy minimization of the system, the initial atomic model of graphite was optimized by the conjugate gradient (CG) method, and the stopping tolerance for energy Δ*E* was 1.0 ×$10^{-10}$. We used the isothermal–isobaric (NPT) ensemble as room temperature (300 K) to update the position and velocity for atoms at each timestep, and we adopted the canonical (NVT) ensemble during compression process. The time step Δt was set at 1 fs. Figure 1a shows the analytical model of graphene in the x−y plane, which was stacked along the *z* direction as shown in Figure 1b. The strain rate in the *x* direction of carbon atoms at x=0 and $x=l_{x}$, marked with the left and right boxes as shown in Figure 1b, was specified as 0.5 Å/ps, and displacements of the same carbon atoms in the *y* direction and *z* direction were fixed. Other carbon atoms were free in each direction, and the *x* and *y* directions were set as periodic boundary conditions.

### 3.2. Mean Curvature

To evaluate the out-of-plane deformation of graphene from plane to curved surface, we proposed a differential geometry method to calculate the mean curvature tensor using the atomistic configuration. During the compression process, each graphene layer, modeled as a 2D plane, initially deformed to a 3D curved surface due to the out-of-plane deformation. Thus, this 2D–3D transition generated folds and kink deformations in the graphite. However, in general, the mean curvature of the differential was used to describe the shape of a folding non-individual body. The structure of graphene is made up of discrete carbon atoms, in which each atom is connected with three adjacent atoms via C–C covalent bonds. As shown in Figure 2, three neighboring atoms I, J, and K are connected to a certain atom O; these four atoms O, I, J, and K span a curved surface *S*. We introduced mean curvature, defined by atomic configuration, so that we could characterize out-of-plane deformation with quantification. It should be sufficient to approximate the normal vector *m* using the normal vector of the plane I–J–K. Here we assumed local Cartesian coordinates O–uvw where the direction of *w* was parallel to the normal vector *m*.

Thus, the curved surface *S* could be expressed as:(1)$$\kappa_{11}u^{2}+2\kappa_{12}uv+\kappa_{22}v^{2}+w=0,$$
where $\kappa_{11}$, $\kappa_{12}$, and $\kappa_{22}$ are the components of curvature tensor κ.
(2)$$\kappa =\left[\begin{array}{cc} \kappa_{11} & \kappa_{12}\\ \kappa_{12} & \kappa_{22} \end{array}\right].$$Equation (Equation 2) corresponds to the first fundamental form of a curved surface in differential geometry.

Since the atom O (u,v,w)=(0,0,0) was located on the surface *S*, we have the following conditions.
(3)$$\begin{array}{cc} & \kappa_{11}u_{I}^{2}+2\kappa_{12}u_{I}v_{I}+\kappa_{22}v_{I}^{2}+w_{I}=0\\ & \kappa_{11}u_{J}^{2}+2\kappa_{12}u_{J}v_{J}+\kappa_{22}v_{J}^{2}+w_{J}=0\\ & \kappa_{11}u_{K}^{2}+2\kappa_{12}u_{K}v_{K}+\kappa_{22}v_{K}^{2}+w_{K}=0. \end{array}$$According to the definition of mean curvature, we have $w_{I}=w_{J}=w_{K}=\bar{w}$, where the constant $\bar{w}$ can be easily calculated using the atomic coordinates of O, I, J, and K. Thus, we calculated the three unknown coefficients, $\kappa_{11}$, $\kappa_{12}$, and $\kappa_{22}$, by solving Equation (Equation 3). To determine the principal direction, the rotation matrix *R* with rotation angle α in the *w* direction was introduced.
(4)$$R=\left[\begin{array}{cc} \cos\alpha & -\sin\alpha \\ \sin\alpha & \cos\alpha \end{array}\right]$$When the two non-diagonal components $\kappa_{12}$ in Equation (Equation 2) vanished, the principal direction $\alpha^{P}$ derived as Equation (Equation 5) helped us to obtain the maximum or the minimum mean curvature.
(5)$$\alpha^{P}=\frac{1}{2}\tan^{-1}\left(\frac{2\kappa_{12}}{\kappa_{11}-\kappa_{22}}\right)$$Using the rotation matrix of *R*, if only the rotation angle in the principle direction was rotated, the curvature tensor $\kappa^{P}$ could be determined from Equation (Equation 6).
(6)$$\begin{array}{cc} & \kappa^{P}=\left[\begin{array}{cc} \kappa_{11}^{P} & 0\\ 0 & \kappa_{22}^{P} \end{array}\right]=R^{T}\kappa R \end{array}$$

The mean curvature *H* is the average value of the principal curvature components $\kappa_{11}^{P}$ and $\kappa_{22}^{P}$, shown as:(7)$$H=\frac{\kappa_{11}^{P}+\kappa_{22}^{P}}{2}$$

## 4. Results and Discussion

### 4.1. Kink Deformation and Delamination

The deformation behavior of graphite under compression depends on the external force and its internal structure. In this section we discuss the effect of compressive strain, the number of graphite layers, and the lattice chirality of the graphite within the layered structure mentioned in Section 2 in detail. The deformation of armchair graphite and zigzag graphite with different numbers of layers of graphene is shown in Figure 3, Figure 4, Figure 5, Figure 6, Figure 7 and Figure 8, from which we could see that most of the graphene layers in each graphite were generated sharply by out-of-plane deformation, similar to folds in strata. The mechanism of kink deformation observed in an Mg alloy was reported in one of our previous works [22]. However, for graphite, not only kink deformation but also delamination appears during the compression process. Furthermore, the types of kink deformation and geometrical properties of delamination strongly depend on the compressive strain, the number of graphene layers, and the direction of compression.

Figure 3 shows the deformation of armchair graphite with 20 graphene layers under compression. After controling the temperature in order to stabilize the structure of the graphite, only microscopic wave-shaped deformations appeared when the compressive strain ε=0.00, as shown in Figure 3a, because of the stability of structure. However, increasing the compressive strain caused the wave-shaped deformations in the *z* direction to become obvious; the mountain and valley shapes could be observed clearly when the compressive strain ε=0.05, as shown in Figure 3b. Moreover, delamination between adjacent layers could be found when the compressive strain ε=0.10, as shown in Figure 3c. As the vdW interaction force between the adjacent layers in the graphite was weaker than that of alloys but stronger than that in ceramics, delamination almost never occurs in alloys with an ong-period stacking ordered (LPSO) phase, but is often observed in ceramics (especially in ceramics with a MAX phase [22]). With the increasing compressive strain, two small pieces of delamination, as shown in Figure 3c, combined to form one large piece of delamination, as shown in Figure 3d, where the valley shapes disappeared.

From the results obtained using armchair graphite with 40 graphene layers under compression, as shown in Figure 4, dependence on each graphene layer in the graphite could be observed more obviously compared to the armchair graphite with 20 graphene layers. When the compressive strain ε=0.10 as shown in Figure 4c, the number of waves in the top layer was totally different from that in the bottom layer. The delamination between the adjacent graphene layers also appeared much later than that in armchair graphite with 20 graphene layers, and only microscopic delamination could be observed. When the compressive strain reached ε=0.25, as shown in Figure 4f, the delamination boundary became more obvious, and two curves consisting of microscopic delaminations could be observed. These two curves were the boundaries of the kink band.

To deeply discuss the effect the number of graphene layers has on micro-buckled graphite, we listed and compared the deformation shapes of armchair graphite with different number of graphene layers when the compressive strain was ε=0.20, as shown in Figure 5. With the increasing number of graphene layers, the delamination between the adjacent graphene layers became smaller and smaller, whereas the boundary of the kink band became more and more obvious. As shown in Figure 5d, the two broken curves indicated the boundaries of the kink band in the armchair graphite with 50 and 60 graphene layers, respectively, where the pieces of microscopic delamination were all on the two kink boundaries; this was because the effective bending rigidity of graphite is a function of the number of layers [33]. We speculated that the fundamental reason for the effects that the number of graphene layers had on the compressive deformation behavior was due to the interlayer distance-dependent interaction of the vdW force.

Figure 6, Figure 7 and Figure 8 show the out-of-plane deformation of zigzag graphite with 20 to 60 graphene layers under compression. Comparing to that of armchair graphite, for the lower numbers of laminated graphene layers, such as *n* = 20, as shown in Figure 3 and Figure 6, when the compressive strain was ε=0.10 and ε=0.15, the delamination in the armchair graphite could more easily be observed than in the zigzag graphite. When the compressive strain reached ε=0.20 and ε=0.25, macroscopic delamination only occurred in one place in the armchair graphite, as shown in Figure 3e,f, whereas delamination appeared in two different places in the zigzag graphite, as shown in Figure 6e,f. Comparing armchair and zigzag graphite with larger numbers of laminated graphene layers when ε=0.20, e.g., *n* = 50, as shown in Figure 5d and Figure 8d, and *n* = 60 as shown in Figure 5e and Figure 8e, only microscopic delamination between adjacent graphene layers appeared. With increasing kink deformation, the boundaries of the kink band passed through microscopic delamination and became more obvious, while most of the pieces of microscopic delamination disappeared. We found that the delamination and the kink deformation simultaneously appeared in graphite, and they controled the growth of each other. As a result, we found that the relationship between kink deformation and delamination in graphite was similar to that of the chicken and the egg, because it was not clear which of the two events was the cause and which was the consequence. As the compression ratio increased, the clear boundaries of the kink band started from the position of delamination and the growth of delamination weakened with the growth of kink deformation. Even in zigzag graphite with 60 graphene layers under compressive strain ε=0.20, as shown in Figure 8e, some carbon atoms jumped out of the graphite due to the growth of the kink deformation.

### 4.2. Compressive Stress–Strain Curves and Average Potential Energy

Figure 9a,b shows the compressive stress–strain curves of armchair and zigzag graphite with 20, 30, 40, 50, and 60 graphene layers, respectively. To clearly see the variation of the stress, we expressed the compressive stress together with the compressive strain from ε=0.00 to ε=0.20. According to the tendency to change of the stress-strain curves, we divided them into three stages. From Figure 9, we could see that all of the stress–strain curves showed a significant peak at the beginning (ε=0.00∼0.01) in the first stage then decreased in the second stage (ε=0.01∼0.03), and increased again in the third stage (approximately ε>0.03). In fact, when the compressive strain reached ε=0.03, out-of-plane deformation occurred in almost all of the graphene layers. Moreover, both armchair and zigzag graphite with 60 graphene layers expressed the maximum compressive stress at ε=0.20, where the kink deformation appeared and enhanced the mechanical behavior of the graphite. On the contrary, both armchair and zigzag graphite with 20 graphene layers showed the least growth in terms of compressive stress due to the generated macroscopic delamination.

Figure 10a,b expresses the relationship between the potential energy per atom and the compressive strain in armchair and zigzag graphite with 20, 30, 40, 50, and 60 graphene layers, respectively. We could see that the potential energy per atom $E_{p}$, in terms of armchair and zigzag graphite with different graphene layers, showed an increasing trend with the increase in compressive strain, where the number of layers and the chirality of graphite had less of an effect on the growth of the average potential energy. The curves were almost monotonically increasing, but for part of them, the increasing ratios were not linear; we speculated that this was because of the appearance of out-of-plane deformation, kink deformation, and delamination in the compressed graphite.

### 4.3. Mean Curvature and Site Potential Energy

According to the derived Equation (Equation 7), we calculated the mean curvature to describe the deformation of the designated graphene layers and discussed the relationship between mean curvature and potential energy in detail. Figure 11, Figure 12, Figure 13 and Figure 14 show the mean curvature and site potential energy of armchair and zigzag graphite with 20 graphene layers under compressive strain ε=0.10 and ε=0.20, respectively, and we have the following: (i) shows site mean curvature, (ii) shows site potential energy, and (iii) shows mean curvature with deformation. The deformation and delamination were in close agreement with Figure 3 and Figure 6, so we could confirm that the site potential energy had a close relationship with mean curvature.

During the compressive process, each graphene layer was generated by out-of-plane deformation. We picked the bottom layer (n=1) and the top layer (n=20), and the layers with large wave deformation due to the inter-layer delamination, as representatives. For instance, when armchair graphite with 20 graphene layers was under compressive strain, ε=0.10 and ε=0.20, delamination was observed between the 11th and 12th layers, as shown in Figure 3, so we expressed the results of the 1st, 11th, 12th, and 20th layers. Mean curvature concentration could be seen from contour maps, which could quantify the drastic change of folds, and a high value of mean curvature indicated a high value of site potential energy.

To discuss the relationship between the mean curvature and site potential energy in detail, we plotted Figure 15. Figure 15 shows the relationship between the mean curvature and site potential energy of armchair graphite with 20 graphene layers under compressive strain ε=0.20, where (a) expresses the relationships of the 1st and 2nd layers, and (b) expresses the relationships of the 11th and 12th layers. As shown in Figure 15a, most of the distribution plot around (0.0, −7.42 eV) and the plus or minus values of the mean curvatures of each site potential energy in the 1st and 2nd layers were nearly the same, which indicated a smaller out-of-plane deformation of the two layers, and there was almost no delamination between the two adjoint layers. From Figure 15b, we could see that although the distributions of the site potential energy of the 11th and 12th layers were concentrated around a point that was almost the same as the one in Figure 15a; however, the site potential energies extended in contrary directions along the white band. Comparing the center of the site potential energy and the opposite curvature distributions of the 11th and 12th layers indicated that obvious delamination was generated between the two layers. Thereby, by evaluating the site potential energy and the mean curvature of each graphene layer in the graphite, we could quantify the kind deformation in the graphite accurately. In addition, evaluation of the site potential energy and the mean curvature was also expected to be used to expose the deformation mechanism of other materials with laminated structures.

## 5. Conclusions

In this work, in order to investigate the interlayer deformation of micro-buckled graphite and to establish a general methodology for evaluating the kink deformation of compressed laminated structures, we performed compression tests on armchair and zigzag graphite with different numbers of laminated graphene layers using MD simulation. We proposed a differential geometric method using mean curvature to explore the relationship between geometrical and mechanical properties. From the compression tests, not only the kink deformation but also the delamination were observed, and different kinds of deformation patterns appeared in the graphite according to the different chiralities of the graphite and different number of graphene layers. We concluded that the kink deformation in graphite started from the position of delamination and the growth of kink deformation weakened the growth of delamination. We also showed the relationships between compressive stress and compressive strain, and between site potential energy and compressive strain to investigate the structural effect on the material properties of graphite. Moreover, according to the calculated mean curvature, we found that the concentration of compressive stress and site potential energy were generated at almost the same locations in all of the analytical models. Lastly, we should note that the calculation of site potential energy and mean curvature using the proposed differential geometric method could be extended and used for solving and predicting the interlayer deformation that occurs in micro and macro materials with laminated layers.

## Figures and Tables

**Figure 1 nanomaterials-12-00903-f001:**
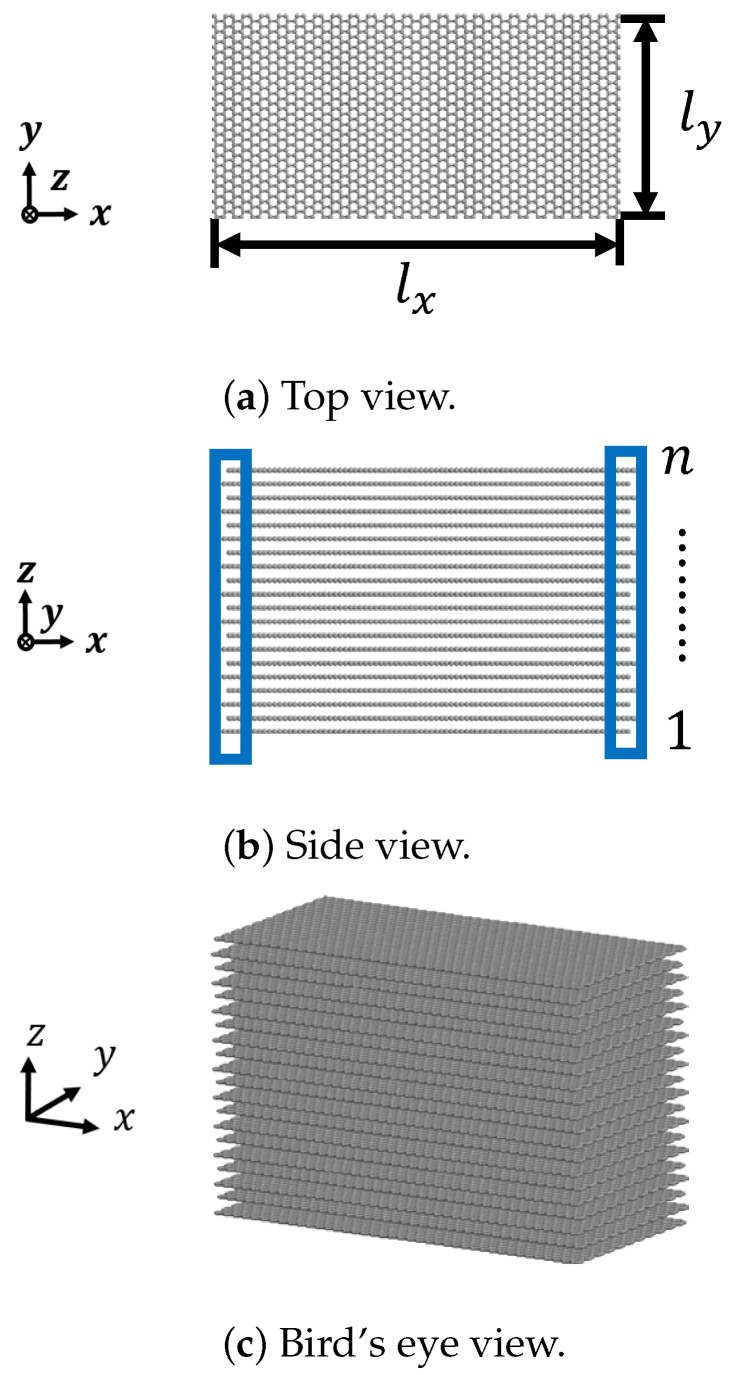
Atomic model of graphite.

**Figure 2 nanomaterials-12-00903-f002:**
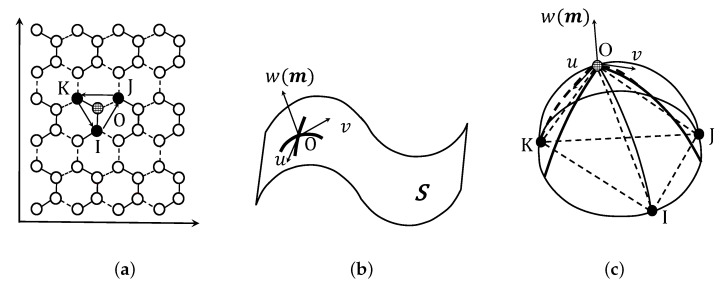
Schematic diagram of mean curvature defined by atomic configuration. (**a**) Discrete lattice structure. (**b**) The approximation of a continuous surface. (**c**) Curvature calculation.

**Figure 3 nanomaterials-12-00903-f003:**
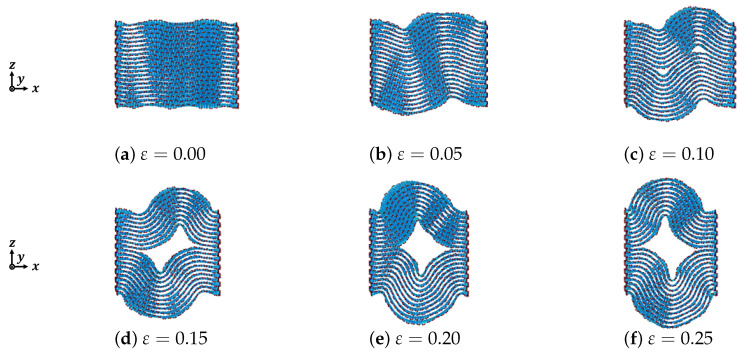
Deformation of armchair graphite under compression (n=20).

**Figure 4 nanomaterials-12-00903-f004:**
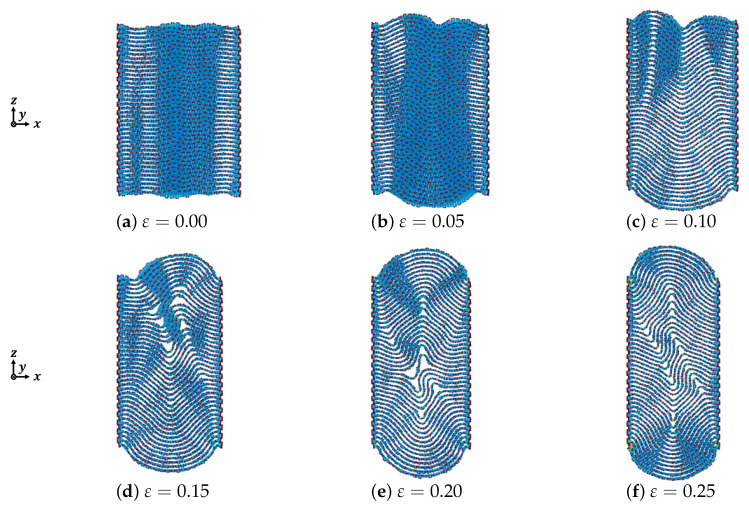
Deformation of armchair graphite under compression (n=40).

**Figure 5 nanomaterials-12-00903-f005:**
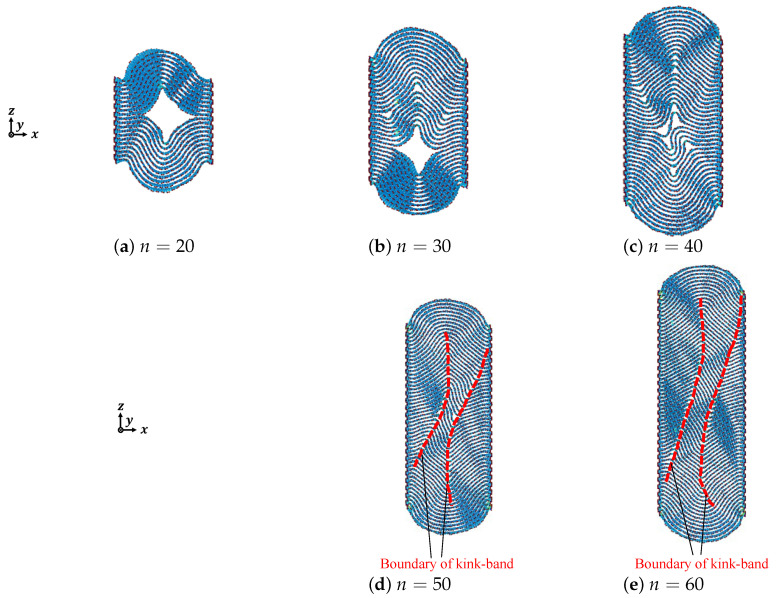
Deformation of armchair graphite under compression (ε=0.20).

**Figure 6 nanomaterials-12-00903-f006:**
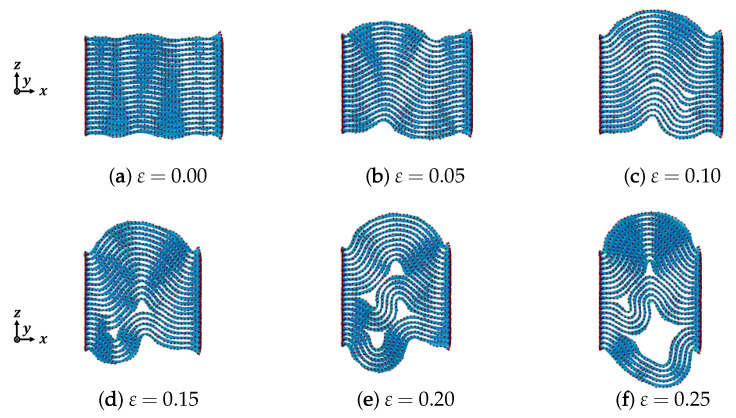
Deformation of zigzag graphite under compression (n=20).

**Figure 7 nanomaterials-12-00903-f007:**
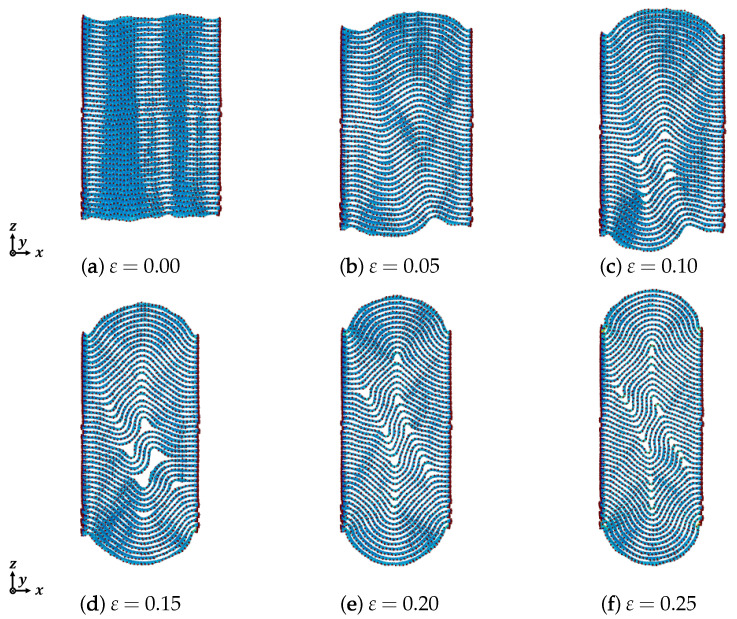
Deformation of zigzag graphite under compression (n=40).

**Figure 8 nanomaterials-12-00903-f008:**
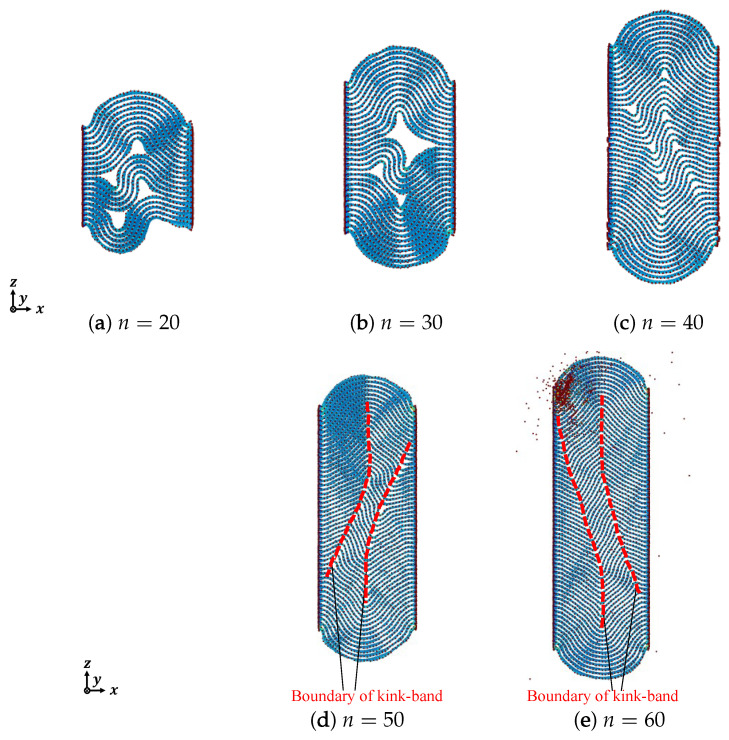
Deformation of zigzag graphite under compression (ε=0.20).

**Figure 9 nanomaterials-12-00903-f009:**
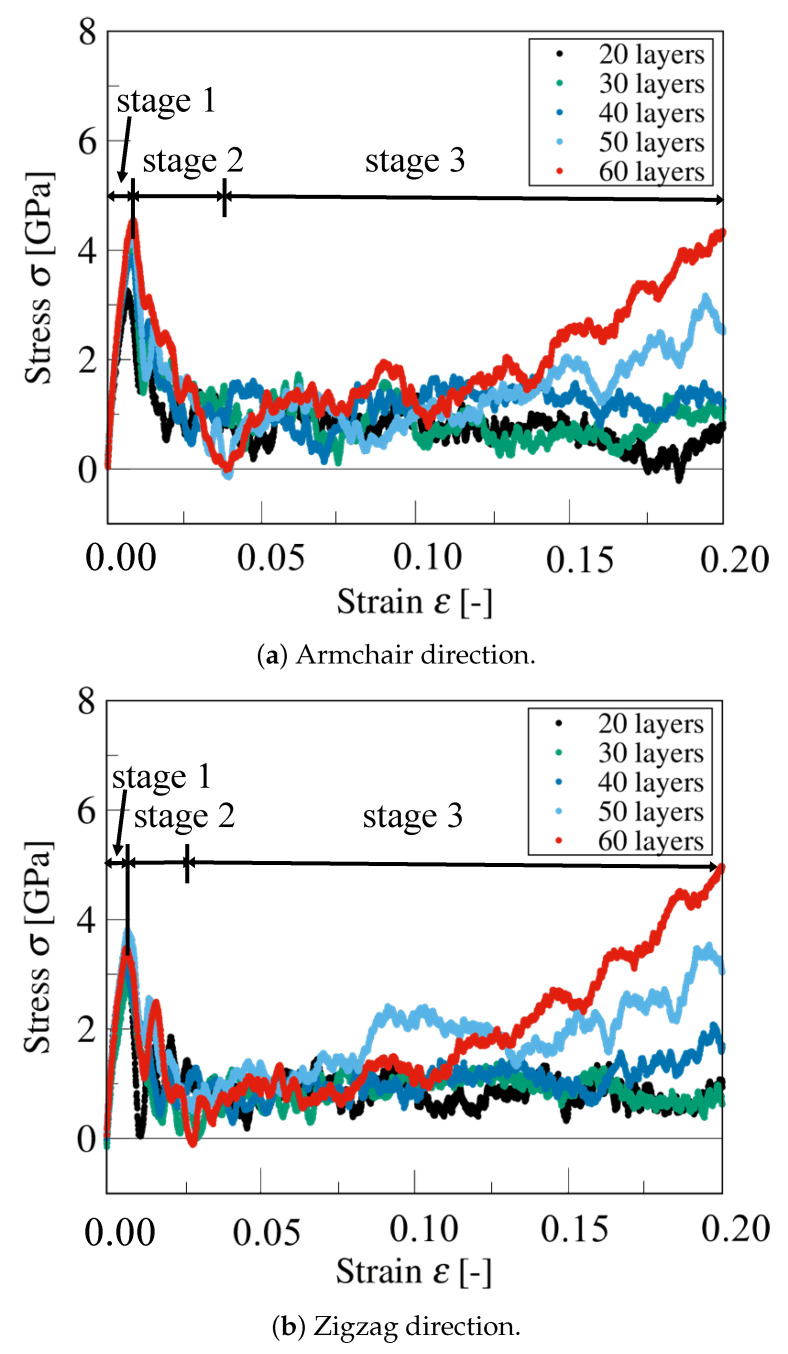
Compressive stress–strain curve of graphite.

**Figure 10 nanomaterials-12-00903-f010:**
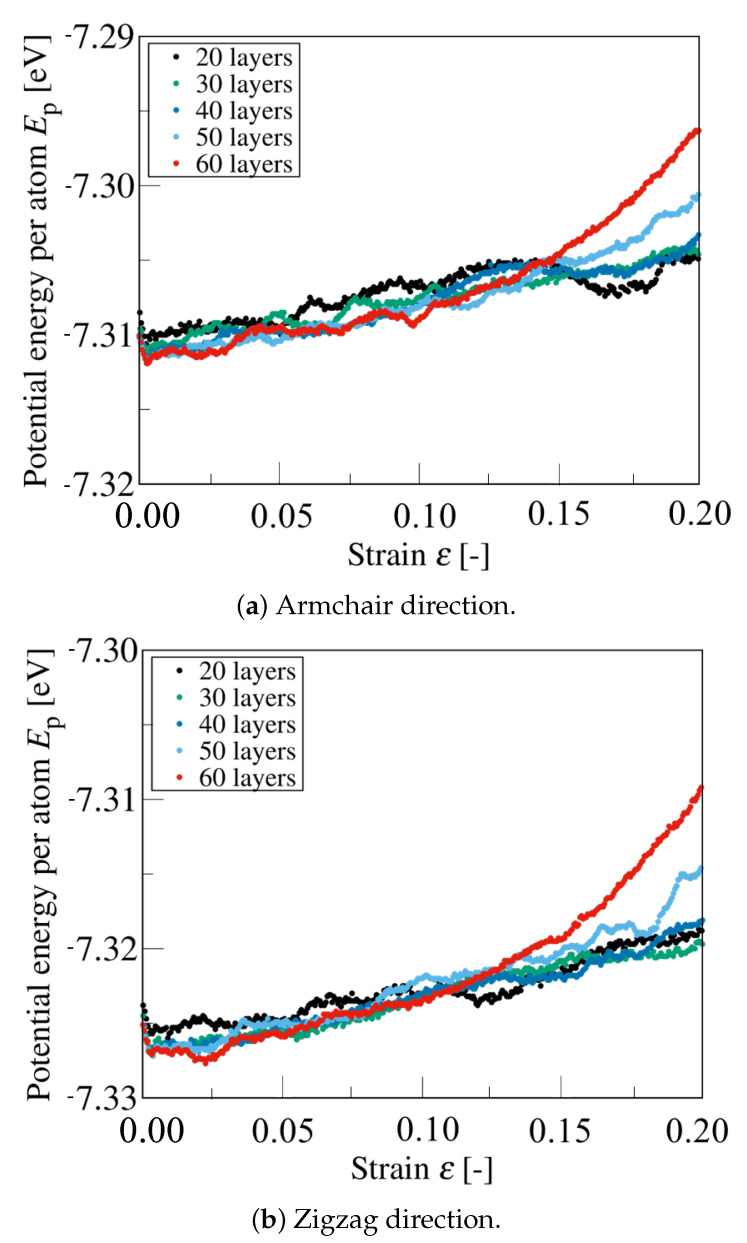
Relationship between potential energy per atom and compressive strain.

**Figure 11 nanomaterials-12-00903-f011:**
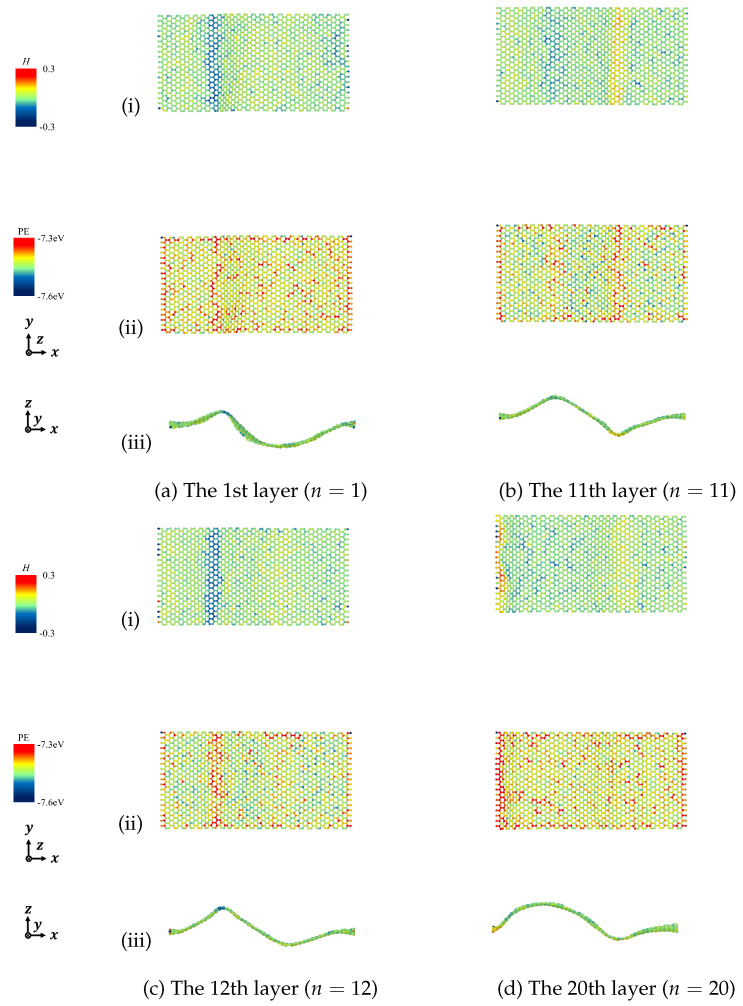
Mean curvature and site potential energy of curved surface of armchair graphite under compression ε=0.10. (i) Site mean curvature, (ii) Site potential energy, and (iii) Mean curvature with deformation.

**Figure 12 nanomaterials-12-00903-f012:**
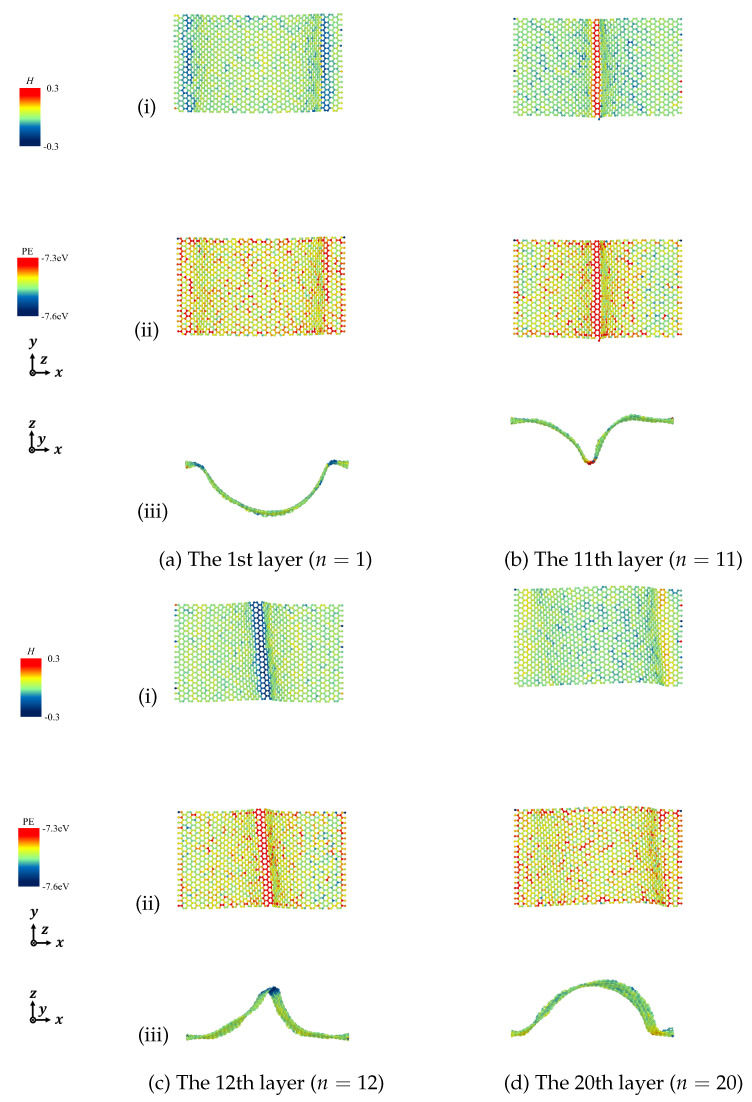
Mean curvature and site potential energy of curved surface of armchair graphite under compression ε=0.20. (i) Site mean curvature, (ii) Site potential energy, and (iii) Mean curvature with deformation.

**Figure 13 nanomaterials-12-00903-f013:**
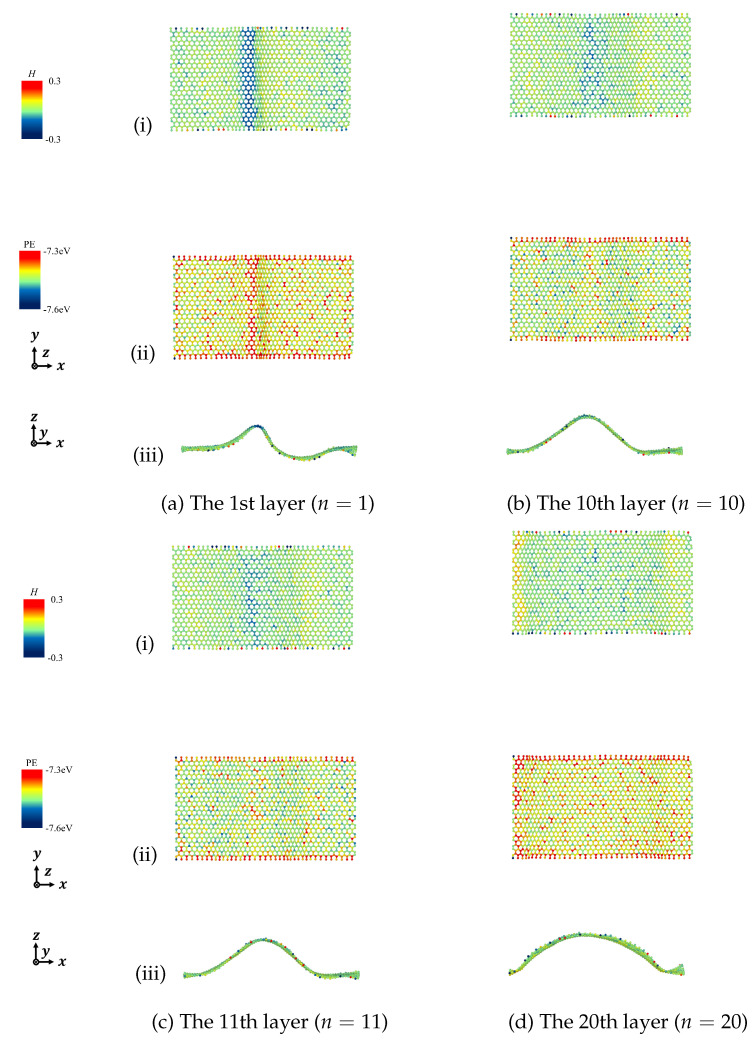
Mean curvature and site potential energy of curved surface of zigzag graphite under compression ε=0.10. (i) Site mean curvature, (ii) Site potential energy, and (iii) Mean curvature with deformation.

**Figure 14 nanomaterials-12-00903-f014:**
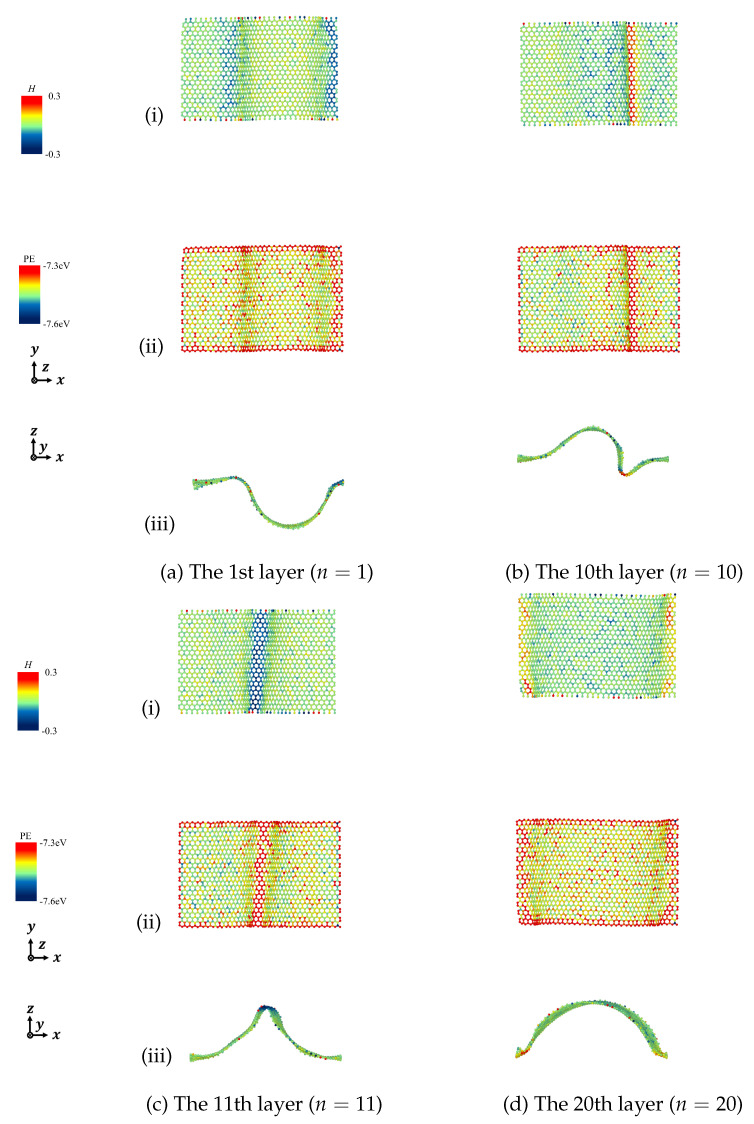
Mean curvature and site potential energy of curved surface of zigzag graphite under compression ε=0.20. (i) Site mean curvature, (ii) Site potential energy, and (iii) Mean curvature with deformation.

**Figure 15 nanomaterials-12-00903-f015:**
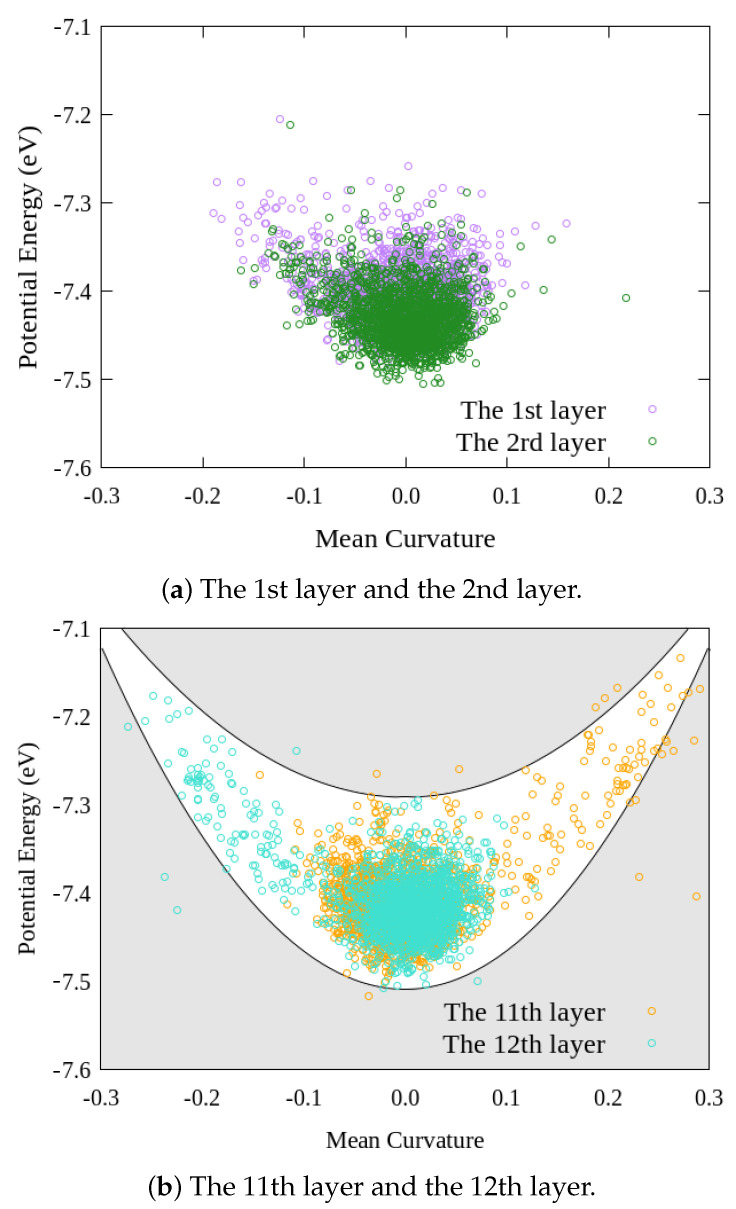
Potential energy and mean curvature of armchair graphite under compression ε=0.20.

## Data Availability

Not applicable.

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
