# Peer review of "The Theoretical Study of Kink Deformation in Graphite Based on Differential Geometric Method"

_nanomaterials, 2022, doi:10.3390/nano12060903_

Round 1
Reviewer 1 Report
This work investigates the interlayer deformation of micro-buckled graphite to establish a general methodology for evaluate the kink deformation of compressed laminated structures, by MD simulation.
The paper is well written and the reader can follow alla steps, I suggest only few changes
- In the Fig. 11, 12, 13 it should be better to use another colour contrast different from cyan/green. The pictures in the upper part of Figs are not so clear.
- In the conclusions it should better underlined and specified the possible applications and implications of obtained results from the practical point of view.
Author Response
We appreciate the encouraging, critical and constructive comments on this manuscript by the reviewer. The comments are very useful for improving the manuscript. We have taken them fully into account in the revised manuscript and believe that the comments and suggestions should enhance the scientific value of the revised manuscript by many folds. The responses are given in attachment.

Reviewer 2 Report
See attached file

Author Response

(The authors gave the same response as above.)

Reviewer 3 Report
I greatly appreciate the subject of the investigations and the simulation methods as well as presented results. The results may explain different experimental observations concerning systems which include multi-layer structures composed of graphene, as in the case of metal matrix composites or electrodeposited metallic layers on the graphene covered substrates. In all this cases the delamination leading to the defragmentation of multi-layer graphene was frequently observed.
The manuscript is written very well, the results of the simulations presented clearly. The general impression is very good. Personally, in the presented context of the paper I would prefer term nano-graphite, not seldom graphite, especially in the title. No important faults in English were found. The figures illustrating the results of the simulations are well chosen and very complete.
In my opinion , the paper should be very interesting for the graphene studying community and may be published in the present form. Congratilations to the Authors.
Author Response
We appreciate the encouraging, critical and constructive comments on this manuscript by the reviewer. The comments are very useful for improving the manuscript. We have taken them fully into account in the revised manuscript and believe that the comments and suggestions should enhance the scientific value of the revised manuscript by many folds. The responses are given in attachment

Reviewer 4 Report
The authors provide us with an interesting investigation by means of molecular dynamics of the buckling of multilayer graphite under compressive strain parallel to the atomic layers. Depending on the number of stacked layers, either microscopic delamination or kink bands are observed at large compression. Stress-strain curves of the compressed structures have been obtained and are discussed in the text, together with local mean curvature and total energy per atom.
A few mandatory remarks to be considered:
Page 6, near line 167: it is not clear on why the graphite equilibrium structure in the absence of strain is not made of (nearly) flat atomic layers. Is the bending wave observed for in Figs 3(a), 4(a), 6(a) and 7(a) solely an effect of temperature? Is it due to the finite number of layers and the boundary conditions applied on the edges along y?
Page 9, near line 221: is it only with zigzag graphite that atoms do escape the structure under large compression? It seems that there are quite a lot of them and only from the layers near the top.
Page 11: please explain how the stress was calculated.
Pages 12 and 13: the text refers to four kinds of drawing illustrated in Fig. 11-14 and labeled as (i), (ii), (iii) and (iv). Actually there are only three drawings in these figures, the contour map of the mean curvature mentioned in Page 12 is not illustrated. Please correct.
Page 13, line 273: it is written "most of the mean curvatures distribute around (0.0, -7.42 eV)". It is not the mean curvature that is distributed but the data in the plane (H,EP) that are distributed around the point with coordinates (0.0,-7.42). Please check the values of the potential energy. In Fig. 10, EP varies in a narrow interval around -7.30 eV/atom, whereas the data in Fig. 15 are dispersed between -7.5 and -7.2. Fig. 10 represents probably an average value but why then does it differ by 0.15 eV between zigzag and armchair models at zero strain?
The English must be improved (see examples below).
Suggestions:
The title should be modified in "Theoretical study OF kink deformation in micro-buckled graphite based on differential geometric method".
Line 84: "atomic models of armchair and zigzag graphite" => Please use a more precise statement, like "we build two graphite models having, respectively, armchair edges and zigzag edges along the x direction. Both models incorporate, 20, 30, 40, 50, or 60 atomic layers"
The matricial equations in eq. (6) can be dropped
Trivial corrections:
Line 80: "establishment of a _generous_ deformation" => Do the author mean a "general"
Line 96: "the height of models _are_ dependent" => correct "are" in "is"
Line 113: correct "interbetween" in "interactions between"
Line 117: please correct "anglethe"
Line 119: ∆E is 1.0E−10 => please provide the units.
Line 121: "atoms in the group" => what do the author mean by "the groups"
Line 197: please correct "Thant is becayse
Line 206: correct "strain reaches _to_ ε = 0.20" in "strain reaches ε = 0.20". Same remark in line 233
Line 246: the expression "are not linearly even minus" is not understandable
Line 250: please explain or correct "crease performance"
Line 270: remove "is" in the sentence "with 20 graphene layers _is_ under"
Line 276: correct "there _has_ almost" in "there is almost"
Reviewer 5 Report
The results obtained in the article «Theoretical study on kink deformation in graphite based on differential geometric method» are novel and worth of publication in Nanomaterials but revision is required.
To my mind, authors should carefully revise the Introduction. There are no References that address the previous attempts of graphite’s deformations mathematical modeling. It’s strange taking into account that such paper were published, for example, https://doi.org/10.1016/j.wear.2021.203652, https://doi.org/10.1103/PhysRevB.92.094108, https://doi.org/10.1080/21663831.2019.1702115, the paper [17] of Barsoum et al. also deserve the mention in Introduction. Some experimental work like https://doi.org/10.1007/s42452-020-2075-y would be also suitable. Author should highlight main results and problems of such paper, to provide the values of obtained mechanical parameters. Then it would be good of authors could compare their results with the mention for verification.
Lines 35-53 can be deleted from the text. Why should we read about deformation in carbon fiber reinforced plastics, MAX phase materials and long-period stacking ordered (LPSO) phases if the object of study is graphite. If it’s important authors should explain why.
Why graphene layers with dimension 102 Å and 49 Å were chosen? How can authors estimate the possible size effect on obtained properties of graphite?
The interesting effect of atoms elimination is observed in Fig.8e. If there any correlations with maps of mean curvature and site potential energy? The eliminated atoms should have the highest potential energy, don’t they?
Round 2
Reviewer 1 Report
All the required changements were addressed
The paper can be accepted in the present form
Reviewer 5 Report
Authors have improved the paper so it can be accepted.